# Diet and the Gut Microbiome as Determinants Modulating Metabolic Outcomes in Young Obese Adults

**DOI:** 10.3390/biomedicines12071601

**Published:** 2024-07-18

**Authors:** Elena N. Livantsova, Georgy E. Leonov, Antonina V. Starodubova, Yurgita R. Varaeva, Aleksey A. Vatlin, Stanislav I. Koshechkin, Tatyana N. Korotkova, Dmitry B. Nikityuk

**Affiliations:** 1Federal Research Center of Nutrition, Biotechnology and Food Safety, 109240 Moscow, Russia; avs.ion@yandex.ru (A.V.S.); varaeva@ion.ru (Y.R.V.); tntisha@gmail.com (T.N.K.); dimitrynik@mail.ru (D.B.N.); 2Therapy Faculty, Pirogov Russian National Research Medical University, 117997 Moscow, Russia; 3Laboratory of Bacterial Genetics, Vavilov Institute of General Genetics Russian Academy of Sciences, 119333 Moscow, Russia; 4Institute of Ecology, Peoples’ Friendship University of Russia (RUDN University), 117198 Moscow, Russia; 5Nobias Technologies, 34 Bld, 123423 Moscow, Russia; st.koshechkin@gmail.com

**Keywords:** obesity, diet, gut microbiome, 16s rRNA sequencing, biomarkers, metabolic syndrome, insulin resistance, dyslipidemia

## Abstract

Obesity, along with metabolic disorders such as dyslipidemia and insulin resistance, increases the risk of cardiovascular disease, diabetes, various cancers, and other non-communicable diseases, thereby contributing to higher mortality rates. The intestinal microbiome plays a crucial role in maintaining homeostasis and influencing human metabolism. This study enrolled 82 young obese individuals, who were stratified into groups with or without metabolic disturbances. No significant differences in the alpha or beta diversity of the microbiota were observed among the groups. Insulin resistance was characterized by an increase in the number of *Adlercreutzia* and *Dialister* as well as a decrease in *Collinsella*, *Coprococcus* and *Clostridiales*. The dyslipidemia and dyslipidemia+insulin resistance groups had no significant differences in the gut microbiota. Dietary patterns also influenced microbial composition, with high protein intake increasing *Leuconostoc* and *Akkermansia*, and high fiber intake boosting *Lactobacillus* and *Streptococcus*. The genus *Erwinia* was associated with increases in visceral fat and serum glucose as well as a decrease in high-density lipoprotein cholesterol. Our findings highlight a significant association between gut microbiota composition and metabolic disturbances in young obese individuals, and they suggest that dietary modifications may promote a healthy microbiome and reduce the risk of developing metabolic disorders.

## 1. Introduction

Obesity and disorders of carbohydrate and lipid metabolism are risk factors for various diseases, including cardiovascular disease, type 2 diabetes mellitus, musculoskeletal disorders and diverse cancer forms, and they are associated with higher mortality rates [1]. The etiology of obesity and associated metabolic disorders is disputable and comprehensive. Overall, diet, lifestyle, genetics, epigenetic modifications, socioeconomic status, and environmental factors have a significant impact on the risk of developing and progressing obesity [2]. Meanwhile, for the past few decades, the place of gut microbiota in the obesity pathogenesis has been receiving growing interest among the scientific community [3]. The intestinal microbiome plays a pivotal role in the maintenance of normal homeostasis and performs a wide range of functions affecting human metabolism. It contributes to the biodegradation of polysaccharides and the extraction of additional energy from food. Additionally, the gut biota integrates into the metabolism of short-chain fatty acids (SCFAs), branched-chain amino acids (BCAAs), bile acids, sulfur-containing amino acids, indole derivatives, trimethylamine N-oxide (TMAO), and vitamins [4,5].

Recent studies suggest that certain issues related to the gut microbiome, such as its composition, diversity index, relative levels, and functional pathways, may predispose to obesity [6]. Mass sequencing techniques (shotgun sequencing, 16s rRNA sequencing) have made it possible to determine the profile of the gut microbiota and how its composition affects hosts’ health, playing a fundamental role in the development of this disease [7].

The normal human gut microbiota comprises predominantly five different phyla: *Bacteroidetes*, *Firmicutes*, *Actinobacteria*, *Proteobacteria*, and *Verrucomicrobia*, with *Bacteroidetes* and *Firmicutes* both accounting for 90% of bacterial species [8]. Obesity is associated with structural and functional alterations in the gut microflora. On the other hand, gut dysbiosis contributes to energy storage and the activation of metabolic pathways, leading to obesity. Traditionally, the gut microbiome in obese individuals is thought to be characterized by an overgrowth of *Firmicutes* and a corresponding decrease in *Bacteroidetes* (*Firmicutes*/*Bacteroidetes* ratio), though contradictory studies exist [9]. Moreover, obese individuals exhibit marked decreases in the genera *Faecalibacterium*, *Oscillibacter*, and Alistipes compared to normal-weight individuals [10]. Specific bacterial species such as *Lactobacillus reuteri*, *Bifidobacterium animalis*, *Methanobrevibacter smithii*, and other Lactobacillus species differ between obese and normal-weight individuals [11]. Current findings highlight the association of the family *Christensenellaceae* with weight loss, which is inversely related to host body mass index (BMI). *Akkermansia muciniphila* is also implicated positively in weight loss [12,13].

The composition of the gut microbiome can be influenced by dietary patterns in different ways. The Mediterranean diet (MedDiet) is characterized by a commitment to the consumption of fresh fruits, vegetables containing fiber, olive oil, red wine, and foods containing polyunsaturated fatty acids such as fish [14]. The gut microbiota of individuals following the MedDiet is characterized by a reduction in the number of representatives of the *Firmicutes* phylum and an increase in the number of *Bacteroidetes*, which is primarily because of the action of fiber-fermenting bacteria involved in the synthesis of SCFAs. At lower taxonomic levels, an increase in the abundance of *Prevotella* and *Lachnospira*, as well as *Roseburia* and *Faecalibacterium prausnitzii*, and a decrease in the abundance of *Ruminococcus gnavus* and *Ruminococcus torques* are observed [15]. A study examining the differences between a vegan diet and a meat-rich diet (MD) revealed that the genus *Coprococcus* was more prevalent in the vegan diet group and less prevalent in MD individuals. In contrast, the genera *Roseburia* and *Faecalibacterium* were observed to be increased in the MD group while being decreased in the vegan diet group [16].

The widespread Western-style diet is characterized by highly processed and refined foods and high levels of sugar, salt, fat, and protein from red meat [17]. The Western diet leads to a decrease in the overall diversity of gut microbes and a shift in the balance of bacterial species. This includes an increase in the *Firmicutes/Bacteroidetes* ratio, often accompanied by higher levels of *Proteobacteria* and a decline in *Bacteroidetes*, notably *Rikenellaceae* and *Prevotellaceae*, as well as a reduction in *Actinobacteria*, especially *Bifidobacterium*. The overconsumption of red meat has been linked to changes in the gut microbiota, such as elevated levels of *Escherichia coli*, *Fusobacterium nucleatum* and *Streptococcus*, and *Bacteroides*. Additionally, a high-salt diet directly diminishes populations of *Lactobacillus* [18].

Dietary fiber intake plays a critical role in maintaining a healthy gut microbiota and is associated with metabolic disorders such as obesity and diabetes [19]. Resistant starch, found in many plant products, can increase the abundance of beneficial bacteria such as *Faecalibacterium prausnitzii* and propionate-producing microorganisms [20]. A low-fiber diet promotes the proliferation of bacteria that degrade the mucus layer of the colon, leading to erosion of the intestinal mucosal barrier and increased susceptibility to pathogens [21]. Meanwhile, excess fat consumption leads to diversity reduction, as high saturated fat intake increases the *Firmicutes*/*Bacteroidetes* ratio, particularly the classes *Mollicutes* and *Clostridiales*, and decreases the abundance of the class *Bacteriodles*. A high-fat diet induces the rise of the abundance of *Shigella*, *Escherichi* and *Enterococcus* at the genus level [22]. Palm oil consumption particularly increases *Verrucomicrobia*, particularly *Akkermansia muciniphila* prevalence [23]. There is a negative correlation between the intake of mono- and polyunsaturated fatty acids and the abundance of *Bifidobacterium*. In contrast, a high-protein diet increases the overgrowth of *Bacteroidetes* while decreasing the relative number of *Actinobacteria* and *Acidobacteria* at the phylum level [24]. 

Most current research considers gut microbiota associations with either diseases or metabolic conditions or with specific diet-patterns, products and nutraceuticals [25,26,27,28]. Meanwhile, the complex relationship between body composition, changes in lipid and carbohydrates metabolism and microbial diversity are still under investigation [29]. Additionally, this sophisticated interconnection is particularly interesting in younger individuals, as the potential targets for early prevention and treatment can be identified. 

This study aimed to determine bacterial biomarkers associated with dyslipidemia and insulin resistance. To meet this aim, a multifactorial analysis of nutritional status in young obese adults, encompassing food diaries, body composition, markers of carbohydrate and lipid metabolism, and the structure of the intestinal microbiome was conducted. Also, the relationship between the consumption of micro and macronutrients and abundance of certain microorganisms was analyzed in order to identify promising targets for dietary intervention. The comprehensive profile generated from such data could offer insights into the health status and future disease risk assessment and aid in selecting more effective therapeutic strategies. 

## 2. Materials and Methods

### 2.1. Subjects and Study Design

A total of 82 Caucasian obese subjects (28 men, 54 women) were examined at the Nutrition Clinic of the Federal Research Center of Nutrition, Biotechnology, and Food Safety. The mean age was 34 years (95% CI: 32.4; 35.9). The mean BMI was 35.80 kg/m^2^ (95% CI: 34.71, 36.97). The mean body weight was 105.02 kg (95% CI: 100.57, 109.48). All participants exhibited low or moderate levels of physical activity as measured by wearable activity trackers.

The distribution of participants by obesity level is presented in Table 1. 

When assessing anthropometric indicators and data body composition (by bioimpedance analysis on InBody 770 analyzer (Inbody Co., Ltd., Seoul, Republic of Korea), a significant increase in waist circumference (WC), hip circumference (HC), fat mass and visceral fat area was noted (Table 2). The average WC was 109.7 cm; the average HC was 112.7 cm; the average fat mass was 45.61 kg; and the mean visceral adipose tissue area was 204.72 cm^2^.

### 2.2. Diet Assessment

We collected 72 h diet dairies from all participants. The nutrients as well as food groups analyses were performed on the NIAP program Scientific Instrument for nutrition analysis (https://nplanner.ru/) based on the food composition databases of Skyrykhin-Tutelyan (http://web.ion.ru/food/FD_tree_grid.aspx, accessed on 15 May 2024), the United States Department of Agriculture (USDA, https://fdc.nal.usda.gov/, accessed on 15 May 2024), etc. [30]. 

### 2.3. Anthropometry, Body Composition and Biochemical Indicators

Body weight and height were measured on a medical scale and stadiometer and performed as kg and m. The BMI was calculated with standard procedures. Body fat mass (kg), muscle mass (kg), fate rate (%), etc. were measured by bioimpedance analysis on an InBody 770 analyzer (Inbody Co., Ltd., Cheonan-si, Republic of Korea). 

Serum total cholesterol (TC), low-density lipoproteins (LDL), high-density lipoproteins (HDL), triglycerides (TG), glucose (Glu), insulin (Ins), aspartate aminotransferase (AST), alanine aminotransferase (ALT), and uric acid levels were analyzed by standard laboratory procedures on a «KONELAB Prime 60i» Laboratory analyzer (Thermo Fisher Scientific, Waltham, MA, USA). Insulin resistance (IR) was calculated with a standard quotation for HOMA-IR (Table 3). 

Based on the serum TC levels, participants were divided into two subgroups: DLD (+) (TC ≥ 5.2) and DLD (−) (TC < 5.2). Depending on the HOMA-IR, participants were distributed into IR (+) (HOMA-IR > 2.7) and IR (−) (HOMA-IR ≤ 2.7) groups. Furthermore, the study population was also separated into DLD-IR (+) and DLD-IR (−) groups (Figure 1). 

### 2.4. Fecal Sample Collection

Patients did not take any sorbents or laxatives (including magnesia and castor oil) prior to sample collection. Fecal samples were collected using sterile collection tubes. The frozen samples were stored at −20 °C and transported to the laboratory on dry ice.

### 2.5. DNA Extraction, Library Construction and Sequencing

Total DNA was extracted using a modified standard method [31]. Briefly, a stool sample was placed in a 2 mL tube to which 0.1 mm and 0.5 mm diameter glass beads (Sigma, Livonia, MI, USA) were added in a 3:1 ratio. Then, 1 mL of warm lysis buffer (60 °C) containing 500 mM NaCl, 50 mM Tris-HCl (pH 8.0), 50 mM EDTA and 4% SDS was added. The mixture was vortexed to obtain homogeneity and homogenized for 3 min using a MiniLys (Bertin Technologies, Rockville, MD, USA). The resulting lysate was incubated at 70 °C for 15 min and then centrifuged at 14,000 rpm for 20 min. Then, 1 mL of the supernatant was transferred to new tubes and placed on ice. Afterwards, 1 mL of lysis buffer was added to the sediment, and the homogenization process was repeated. The supernatants were combined in 15 mL tubes and supplemented with 4 mL of 96% ethanol and 200 mL of 3 M sodium acetate. The mixture was incubated at −20 °C for at least one hour and then centrifuged at 14,000 rpm at +4 °C for 15 min. The resulting precipitate was washed twice with 80% ethanol, dried at 53 °C for 30–60 min, and dissolved in 200 mL of sterilized MilliQ water. The solution was centrifuged again and transferred to new tubes. RNase A (5 mg/mL) was added to the solution and incubated at 37 °C for 1 h. Chloroform was added to the solution at a 1:1 ratio, mixed by vortexing for 1 min, and centrifuged at 5000× *g* for 5 min. The liquid phase was carefully transferred to a new sterile tube and used to perform the PCR. The resulting DNA solution was stored at −20 °C.

Amplification of the V4 variable region of the 16S rRNA gene was carried out in one round using a Verity thermal cycler (Applied Biosystems, Waltham, MA, USA). PCR products were purified using a Cleanup Mini DNA isolation kit for reaction mixtures (Evrogen, Moscow, Russia). The concentration of the resulting 16S libraries in solution was determined using a Qubit^®^ fluorimeter (Invitrogen, Waltham, MA, USA) with the Quant-iT™ dsDNA High-Sensitivity Assay Kit. Purified amplicons were mixed equimolarly based on the obtained concentrations. The quality of the library prepared for sequencing was assessed using agarose gel electrophoresis.

Further sample preparation and sequencing of the pooled sample were conducted using the MiSeq Reagent Kit v2 (500 cycles) and the MiSeq instrument (Illumina, San Diego, CA, USA) according to the manufacturer’s recommendations. After quality score trimming, DNA read pooling was performed using the SeqPrep package v. 1.2, resulting in a final read length of 252 bp.

### 2.6. Gut Microbiota Composition Profiling

For each sample, at least 5000 reads were obtained per sample. During the sequencing process, a negative control sample was sequenced along with samples from each batch; reads with a high probability of corresponding to contaminating bacteria found in the negative control were removed from the real sample files. Next, the data were analyzed using the analytical system Knomics-Biota (https://biota.knomics.ru/, accessed on 15 May 2024), including basic filtering and assessment of data quality, profiling of taxonomic composition, visualization, and comparison of composition with meta-data [32]. The main steps of the analysis are briefly described below.

Raw reads were preprocessed using the QIIME v. 1.9.2 software package [33]. Low-quality ends were trimmed using the split_libraries_fastq.py function (with a quality threshold of --phred_quality_threshold 19), and reads for which the trimmed length was less than 75 percent of the original length were discarded from further analysis. Next, the reads were mapped onto the GreenGenes 13.5 database (with a 97% degree of similarity between taxa) using the pick_closed_reference_otus.py function of the QIIME v. 1.9.2 program, as a result of which tables of representation of relative taxonomic units (OTUs) were obtained [34]. Further analysis included samples for which at this stage there were at least 5000 reads. Alpha diversity was assessed after thinning the resulting tables of OTU representation to 5000 reads per sample using the Shannon and Chao1 metrics. Tables of representation at the levels of species, genus, family, etc. were obtained by summing the representation of OTUs belonging to the corresponding taxonomic group. The metabolic potential of the microbial community was assessed using the PICRUSt2 program v. 2.3 [35].

In addition, to validate the results, an algorithm was applied that allows the community composition to be assessed at a more detailed level compared to OTUs. This approach consisted of applying the DADA2 algorithm to preprocessed (after trimming low-quality ends) reads to obtain representative sequences [36]. Next, the taxonomic classification of these sequences was carried out using a classifier implemented in the QIIME 2 (2020.6) software package and trained on the GreenGenes 13.5 database [33,37]. These sequences were trimmed in accordance with the primers using the TaxMan program and aggregated to obtain 97% similarity between taxa using CD-hit [38,39].

### 2.7. Statistical Analysis

We analyzed participants’ characteristics using the Statistical Package for the Social Sciences software version 20.0 (SPSS Inc., Chicago, IL, USA). The parameters investigated were expressed as mean and standard deviation (SD) for parametric distributions or as median value with 25th and 75th percentiles for nonparametric distributions. Significance tests of alpha-diversity indices (Shannon and Chao1) were conducted using the Wilcoxon test. Principal coordinates analysis (PCoA) based on Unifrac distances at the genus level was utilized for beta-diversity to visualize differences in the microbial community structure across samples. Principal component regression (PCR) was performed for beta diversity. We also performed a differential population analysis based on the results of the gut microbiota composition profiling using the permutation test. Statistically significant taxa were selected (*p*-value < 0.05). log2FC was calculated for each statistically significantly different taxon, where the fold change is the ratio of the average number in patients with and without metabolic disorders. log2FC was used for the convenience of presenting the results.

## 3. Results

### 3.1. Characteristics of Carbohydrate and Lipid Metabolism

According to the biochemical blood test, there was an increase in the level of transaminases (AST, ALT) in 5 (6.1%) and 20 (24.4%) patients, respectively: AST in 3 men and 2 women, and ALT in 12 men and 8 women. Uric acid was elevated in 38 (46.3%): 23 men and 15 women. The average AST level was 22.65 U/L; ALT—29.45 U/L, uric acid—356.61 μmol/L. An increase in total cholesterol level (TC) was detected in 44 (53.7%) patients: 20 men and 24 women, with an average TC level of 5.19 mmol/L. An increase in the level of low-density lipoprotein (LDL) cholesterol was detected in 33 (40.2%). The initial biochemical parameters and lipid spectrum are presented below in Table 3. An increase in the HOMA index was detected in 40 (48.8%) patients: 17 men and 23 women. The average fasting blood glucose level was 5.05 mmol/L; insulin—14.35 μIU/mL. The average HOMA index was 3.26 (Table 3).

### 3.2. Diet

According to food consumption analysis (72-h food record), the diet of participants before was characterized by a high consumption of proteins, fats, including saturated (SFA) and unsaturated (UFA) fatty acids, *n*-6 PUFAs (n3:n6 ratio is 1 to 8), mono- and disaccharides, sodium and an insufficient consumption of vitamins B9, D, and beta-carotene when comparing actual nutrition data with the physiological needs for energy and nutrients (Appendix A) [30].

The groups were comparable in the energy intake: 2825.1 ± 1141.3 vs. 3118.1 ± 1513.9 kcal/d for the DLD (−) and DLD (+) groups, respectively; 3077.8 ± 1601.8 vs. 2888.4 ± 1059.9 kcal/d for the IR (−) and IR (+) groups, respectively; and 3050.6 ± 1465.2 vs. 2816.9 ± 1042.2 kcal/d for the DLD-IR (−) and DLD-IR (+) cohorts, respectively. Macronutrient intake also did not differ significantly between the study groups, with patients in the DLD and IR cohorts having a comparably higher intake of SFA, and the DLD-IR cohort performing a tendency to a lower dietary fiber consumption. In addition, the results showed reliable differences in some nutrient intake, with higher levels of omega-6 and zinc in participants with hyperlipidemia, and lower levels of beta-carotene, tocopherol, resistant starch, pectin and oxalic acid in the insulin-resistant group. The DLD-IR cohort had lower intakes of pectin, resistant starch and phytosterols (Figure 2).

### 3.3. Association of Gut Microbiota Composition and Dietary Variation with Clinical Parameters and Metabolism

The composition of the gut microbiota was analyzed using 16s rRNA sequencing. In total, 29 phyla, 481 genera, and 586 species of microorganisms were identified. The most abundant phyla were *Firmicutes* (86.3 ± 7.9%), *Actinobacteriota* (6.4 ± 5.5%), *Bacteroidetes* (2.9 ± 4.5%), *Proteobacteria* (2.1 ± 4.1%), and *Verrucomicrobiota* (1.0 ± 3.5%). However, the insulin-resistant cohort exhibited elevated levels of *Proteobacteria* (2.6% vs. 1.7%) and reduced *Verrucomicrobiota* (0.5% vs. 1.2%) compared to their non-insulin-resistant counterparts; similar indicators were observed in the DLD-IR groups (3.3% vs. 1.7% and 0.4% vs. 1.1% for *Proteobacteria* and *Verrucomicrobiota*, respectively). In addition, *Bacteroidetes* were less abundant in the DLD-IR group (1.9% vs. 3.2%). At the genus level, predominant representatives included unclassified *Clostridiales* (12.4 ± 5.1%), unclassified *Ruminococcaceae* (12.0 ± 5.8%), unclassified *Lachnospiraceae* (10.4 ± 3.7%), *Blautia* (12.7 ± 6.9%), *Feacalibacterium* (5.6 ± 5.3%), *Coprococus* (4.9 ± 2.5%), and *Bifidobacterium* (3.8 ± 5.2%). The *Firmicutes*/*Bacteroidetes* ratio was high in all presented cohorts but did not have significant differences. A detailed graphical representation of the data is provided in Figure 3.

We analyzed differences in microbial community structures among study groups, measuring alpha diversity using the Chao1 and Shannon indices. There were no reliable differences in alpha diversity between groups but there was a trend toward decreased diversity in the IR (+) and DLD-IR (+) cohorts (Figure 4a,b). The *Firmicutes*/*Bacteroidetes* ratio was high in all presented groups but did not have significant differences. Beta diversity was estimated using unweighted and weighted Unifrac distances analyzed by principal coordinate analysis (PCoA). There were also no differences in beta diversity between the study groups (Figure 4c).

Significant intergroup changes were identified at the genus and species level. Results showed that participants with dyslipidemia had no significant difference in levels of gut microbiota compared with the participants without dyslipidemia. Furthermore, no significant difference was observed between the DLD-IR (+/−) groups. Participants with impaired carbohydrate metabolism were characterized by an increase in the number of *Adlercreutzia* and *Dialister* with a decrease in *Collinsella aerofaciens*, *Coprococcus* and *Clostridiales*. The data are presented in Figure 4d.

Furthermore, we examined the relationship between the composition of the microbiome, body composition and key biochemical indicators of carbohydrate and lipid metabolism. The results demonstrate that representatives of *Lactobacillus*, *Dialister*, and *Veillonellaceae* are positively associated with weight and visceral fat mass. *Veillonellaceae*, *Dialister*, *Erwina*, *Lachnispira*, families S24-7, *Enterobacteriaceae*, and the order *Acitomycetales* were associated with lower BMI. And *Blautia obeum*, *Coprococcus*, and *Erysipelotrichaceae* were correlated with lower visceral fat in obese adults.

At the phylum level, *Acinobacteria* was negatively associated with TG levels, while *Bacteroides* was associated with higher serum HDL abundance. *Proteobacteria* was correlated with lower HDL and high ApoB, and *Verrucomicrobia* abundance was negatively associated with TC, LDL, TG, and ApoB levels. The genus *Bifidumbacteriaum*, *Bactroides uniformis* and *Faecalibacterium prausnitzii* were correlated with lower blood glucose levels, while *Erwinia*, *Lactococcus* and *Akkermansia muciniphila* were associated with higher glucose levels. Moreover, lower levels of TC and LDL were associated with Lactobacillus, *Streptococus* and *Akkermansia muciniphila*. The data are presented in Figure 5.

To assess the contribution of dietary macronutrients to the composition of the gut microbiota at the genus and species levels, a Pearson correlation analysis was performed between microbial abundance and nutrient intake for the entire study sample. The results showed that higher protein intake leads to a rise of *Leuconostoc*, *Holdemania* and the genus *Akkermansia*, while a decrease in *Butyricimonas* is noted. *Leuconostoc* was positively and *Prevotella* was negatively correlated with SFA consumption. High mono- and disaccharides intake is associated with increased levels of *Lactobacillus* zeae, *Coprococcus* spp., p-1630-c5 and *Acinetobacter lwoffii* and decreased levels of *Mogibacterium* and *Ruminococcus*. High fiber intake reduces the abundance of *Oscillospira* and increases *Lactobacillus zeae*, *Streptococcus anginosus*, *Coproccocus*, *Bulleidia*, *Holdemania*, *Neisseria*, *Cardibacterium*, *Acinetobacter lwoffii*, and *Akkermansia*. The data are presented in Figure 6a,b.

## 4. Discussion

Despite extensive research into the relationship between obesity, metabolic disorders and the gut microbiome, numerous aspects remain unclear. The impact of different diet patterns and nutrients on the composition of the intestinal microbiome and the potential for its dietary correction are of particular interest.

This study aimed to investigate dietary patterns, biochemical markers of carbohydrate and lipid metabolism, and gut microbiota composition in young adults with dyslipidemia, insulin resistance, and a combination of both.

The results demonstrated that there were no significant differences between the groups in alpha diversity, which was evaluated using the Shannon and Chao1 indexes. Beta diversity was also not significantly different. Current understanding suggests an association between obesity and metabolic disorders with reduced gut microbiota diversity; however, evidence is controversial. For instance, in a study of obese women with and without metabolic syndrome, an increase in alpha diversity was observed in the group with both obesity and metabolic syndrome [40]. Another study showed a reduction in diversity in groups with obesity but not in those with diabetes [41]. Furthermore, research conducted by Sroka-Oleksiak et al. revealed that there was no marked difference in either alpha or beta diversity between participants with obesity, diabetes, and healthy controls [42].

Meanwhile, reliable differences in the genus and species diversity of microorganisms were observed among the groups. The genus *Dialister* was more abundant in the IR (+) group. Furthermore, *Dialister* was associated with higher body weight but lower plasma triglyceride concentrations. *Dialister* is a member of the *Veillonellaceae* family, and it demonstrated a decreased abundance in insulin-sensitive obese and overweight adults [43]. In another study, a higher abundance of the genus *Dialister* was associated with impaired glucose tolerance in overweight or obese African-American men [44]. Moreover, there is evidence indicating a positive correlation between the presence of *Dialister* and the visceral fat area [45]

The abundance of *Adlercreutzia* was higher in the group with impaired carbohydrate metabolism. Additionally, a long-term study has shown that *Adlercreutzia* is associated with increased BMI [46]. The high abundance of Adlercreutzia was associated with a high-fat, high-sugar diet in a study of C57BL/6J mice [47]. However, there is evidence that the amount of *Adlercreutzia* in the intestine is lower in individuals with non-alcoholic fatty liver disease [48]. *Collinsella aerofaciens* was reduced in the group of participants with insulin resistance. In a study of overweight and obese pregnant women, low dietary fiber intake was associated with an increase in *Collinsella* abundance [49]. There is also evidence that *Colinsella* overgrowth is associated with a number of inflammatory diseases, such as rheumatoid arthritis [50].

The present study demonstrated a correlation between an increased abundance of *Blautia obeum* and a higher visceral fat area as well as a decreased concentration of ApoB. A number of studies have demonstrated a correlation between the growth of *Blautia obeum* and obesity as well as a negative impact on various parameters, including weight, BMI, and visceral fat [51,52]. However, the data on the effect of diet on the abundance of *Blautia obeum* remain debatable. A study conducted among women in South Korea found that *Blautia obeum* was associated with a diet with a low glycemic index. Another study of 1425 individuals in the Netherlands found a significant correlation between the genus *Blautia* and a diet characterized by the frequent consumption of fast food [53]. 

*Coprococcus* was associated with a higher intake of dietary fiber and lower triglyceride levels, and its abundance was decreased in the insulin-resistant group. In a study of 353 individuals in the American population, *Coprococcus* abundance was associated with better insulin sensitivity [54]. The vegan diet was shown to promote the growth of *Coprococcus*, while the meat-rich diet suppressed it [16]

The genus *Holdemania* (family *Erysipelotrichaceae*) was correlated with a high-protein and high-fat diet according to the study. Members of this genus are capable of fermenting simple sugars, including fructose, glucose, and sucrose. However, they are unable to ferment complex carbohydrates, such as polyols and starch. A reduction in the abundance of *Holdemania* was found to be associated with an improvement in glucose tolerance test scores [55]. The species of the genus *Ruminococcus* showed a negative correlation with mono- and disaccharide intake. A study conducted on 46 obese adult subjects revealed a negative association between *Ruminococcus* abundance and insulin resistance, as measured by the HOMA-IR index [56]. 

The genus *Erwinia* was correlated with body composition, lipid and carbohydrate metabolism measures but was not associated with macronutrient intake. The growth of genus members is associated with an increase in visceral fat, serum glucose and ApoB levels and a decrease in HDL and ApoA levels consequently. According to scientific literature, the health effects of this bacterium are inconclusive: in some cases, low levels of *Erwinia* have been associated with the development of diabetic retinopathy, while another study demonstrated its association with hypertension and increased salt intake [57,58].

*Bifidobacterium*, particularly *Bifidobacterium adolescentis*, was linked to decreased plasma triglyceride and glucose levels as well as increased dietary fat intake. Bifidobacterium is known for its positive health effects, such as reduced BMI, dyslipidemia, and hypertension, possibly through an increased production of short-chain fatty acids [59,60].

A positive association was observed between the abundance of *Akkermansia* and lipid metabolism, although an increased glucose concentration was also reported. A diet with an increased intake of protein, fat, and fiber contributed to the increased abundance of this bacterium. *Akkermansia muciniphila* is now regarded as a representative member of the gut microbiome with a positive effect on weight loss and a greater prevalence in individuals without obesity [61].

### Study Limitations

The total number of participants was limited, and there were not enough male subjects to conduct additional subgroup analyses by gender. This study provides evidence for differences in the gut microbiome structure but does not include the entire species composition for analysis. This limitation is because of the use of a single region (v4) for 16s rRNA sequencing, which allows the identification of some bacterial species with reliable accuracy [62]. In this study, we used an OTU-based approach to analyze microbiota data, which has lower resolution compared to ASV-based analysis. The Pearson correlation used in our work may have a number of disadvantages compared to specialized statistical approaches, whereas this test has been widely used for such studies. Although investigating the microbiome of healthy lean individuals was not the aim of our study, a comparison of results with this group may be useful in identifying obesity features. Further large-scale longitudinal studies are required to clarify the relationship between different metabolic conditions and microbiome structure.

## 5. Conclusions

Obesity and metabolic disorders are common health conditions and tend to increase in prevalence worldwide. Early diagnosis using evidence-based biomarkers can help in the prevention of these disturbances. The gut microbiome composition represents a promising target for the treatment of obesity-associated diseases. Discovering opportunities to modify the intestinal microbiome through dietary interventions and the use of probiotics is a relevant target of current research in this field. Overall, our study established an association between features of gut microbiome composition, dyslipidemia, insulin resistance and dietary patterns in young obese adults. The development of approaches to the dietary management of metabolic disorders through specific effects on the structure of the gut microbiome may significantly improve current strategies for the treatment of various diseases. Particularly, increasing the intake of dietary fiber and non-digestible starch, as well as a more diverse micronutrient composition of foods, can promote a healthy microbiome and reduce the risk of developing metabolic disturbances. Further studies are needed to expand the knowledge of the relationship between metabolic disorders and the gut microbiome and to reveal mechanisms and pathways that will allow the development of new treatments and prevention approaches.

## Figures and Tables

**Figure 1 biomedicines-12-01601-f001:**
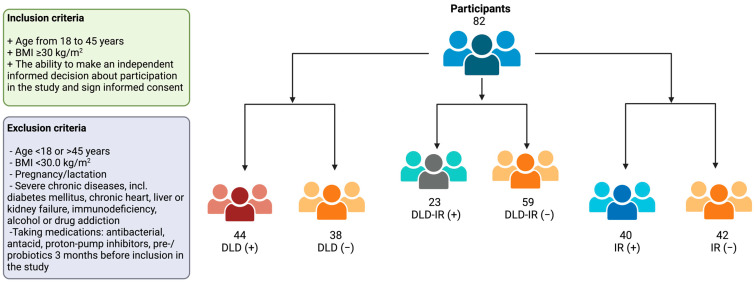
Flowchart visualizing participant recruitment. A total of 82 participants were enrolled in the study. Individuals were stratified into groups with or without dyslipidemia (DLD), insulin resistance (IR), and the combination (DLD-IR).

**Figure 2 biomedicines-12-01601-f002:**
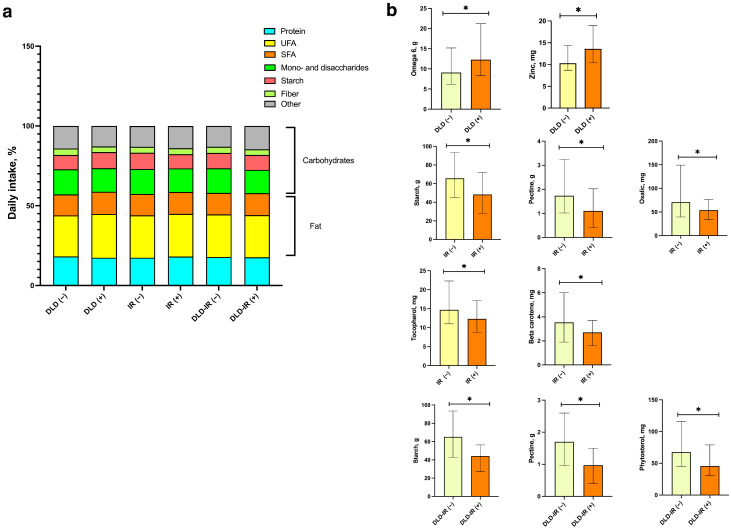
Summary of relative daily macronutrient contribution (% energy) in the study groups (**a**). Measured differences in daily intake (g/day) of nutrients among different groups. Data are presented as median and interquartile range (**b**). * *p* ≤ 0.05.

**Figure 3 biomedicines-12-01601-f003:**
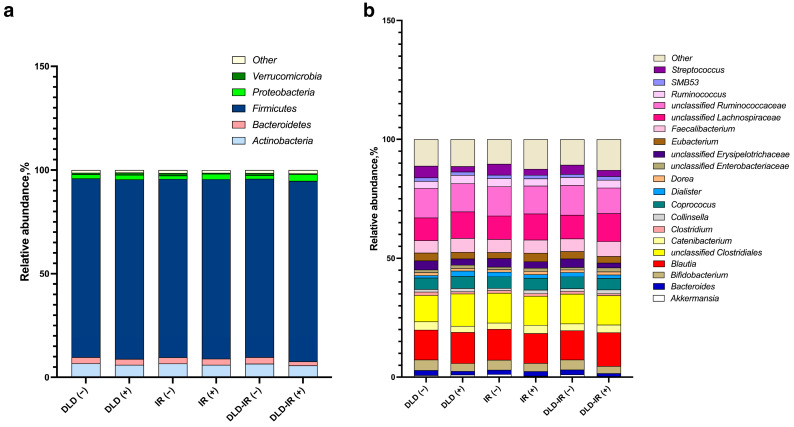
Gut microbiota composition. The average relative abundance of the 5 most abundant phyla showed a high prevalence of *Firmicutes* and a reduced number of the remaining phyla (**a**). The composition of the average relative abundance of the 20 most represented genera (**b**).

**Figure 4 biomedicines-12-01601-f004:**
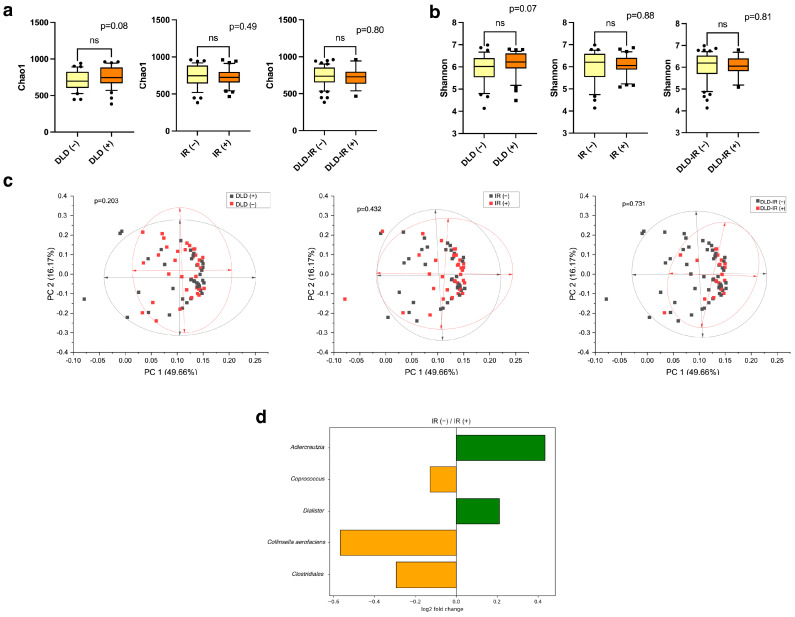
Comparative analysis of gut microbiota in the DLD (+/−), IR (+/−) and DLD-IR (+/−) groups. Alpha diversity was assessed using several metrics: Chao1 index (**a**), Shannon index (**b**) (ns: not significant). Beta diversity of bacteria identified with the principal coordinates analysis (PCoA) using Unifrac distances measures (**c**). To assess statistical differences between groups, a permutation test was used and log2FC was calculated. log2FC < 0 means that this taxon is more presented in IR (+) patients, log2FC > 0 means that the taxon is more presented in IR (−) patients. Significant intergroup changes were identified at the genus and species level. log2FC was calculated for each significantly different taxon (*p*-value < 0.05) (**d**).

**Figure 5 biomedicines-12-01601-f005:**
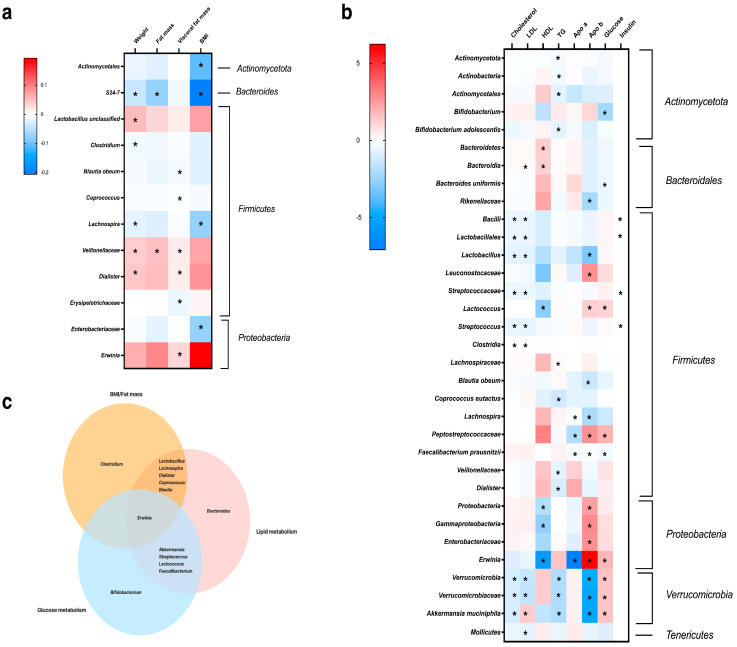
Heatmap analysis of association between gut bacterial taxa and body composition (**a**), serum glucose and lipids (**b**). Interactions between bacteria at the genus level associated with changes in BMI/fat mass, lipids, and glucose metabolism (**c**). * *p* ≤ 0.05.

**Figure 6 biomedicines-12-01601-f006:**
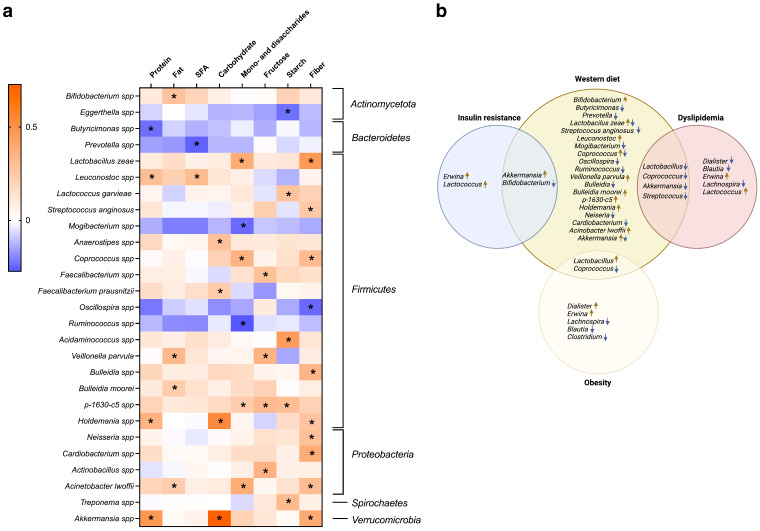
Pearson correlation between the relative abundance of gut bacteria (at the genus and species level) and dietary macronutrient intake (**a**). Crossover between microorganisms that have been associated with the Western-style diet, insulin resistance, dyslipidemia and obesity in this study (**b**). * *p* ≤ 0.05.

**Table 1 biomedicines-12-01601-t001:** Demographics, obesity class distribution, and prevalence of metabolic disorders (*n* = 82).

Parameter	Mean (95% CI)/*n* (%)
Age	34.0 (32.4; 35.9)
Gender	
male	28 (34.1)
female	54 (65.9)
Obesity	
Class I	41 (50.0)
Class II	25 (30.5)
Class III	16 (19.5)
Smoking status	
Current smokers	22 (26.8)
Non-smokers	60 (73.2)
Dyslipidemia	44 (53.7)
Insulin resistance	40 (48.8)
Hyperuricemia	38 (46.3)
Hypertension	24 (29.3)

**Table 2 biomedicines-12-01601-t002:** Anthropometry and body composition analysis (*n* = 82).

Parameter	Mean (95% CI)
Body weight, kg	105.02 (100.57; 109.48)
BMI, kg/m^2^	35.80 (34.71; 36.97)
Waist circumference, cm	109.7 (107.1; 112.3)
Fat mass, kg	45.61 (43.02; 48.20)
Visceral adipose tissue area, cm^2^	204.72 (195.45; 213.98)
Muscle mass, kg	33.53 (31.72; 35.34)
Total body water, L	43.55 (41.36; 45.74)
Extracellular fluid, L	16,46 (15.64; 17.28)

**Table 3 biomedicines-12-01601-t003:** Biochemical parameters and lipid profile (*n* = 82).

Parameter	Mean (95% CI)	Reference
Glucose, mmol/L	5.05 (4.94; 5.15)	3.9–5.6
AST, U/L	22.65 (20.40; 24.90)	8–33
ALT, U/L	29.45 (24.72; 34.18)	7–56
Uric acid, μmol/L	356.61 (338.28; 374.94)	M 110–420; W 110–360
TC, mmol/L	5.19 (4.98; 5.41)	3.5–5.2
TG, mmol/L	1.49 (1.31; 1.68)	0.68–1.69
LDL cholesterol, mmol/L	3.43 (3.24; 3.62)	<3.2
HDL cholesterol, mmol/L	1.17 (1.13; 1.22)	0.9–1.2
Insulin, μIU/mL	14.35 (12.43; 16.26)	2.6–24.9
HOMA index	3.26 (2.79; 3.73)	≤2.7

## Data Availability

The data presented in this study are available on request from the corresponding author. The sequencing data were submitted to the NCBI SRA repository under the accession number PRJNA1131598.

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
