# Peer review of "Diet and the Gut Microbiome as Determinants Modulating Metabolic Outcomes in Young Obese Adults"

_biomedicines, 2024, doi:10.3390/biomedicines12071601_

Round 1

Reviewer 1 Report

Comments and Suggestions for Authors

There are several areas where the manuscript should be improved:

1. Highlight the implications or potential applications of the study’s findings in the abstract section.

2. The review of literature could be expanded to include more diverse studies, particularly those exploring different dietary patterns and their specific impacts on the gut microbiome. Include a broader range of literature to provide a more comprehensive background.

3. The research objectives could be stated more clearly. While the introduction touches upon various aspects, a succinct statement of the study's aim would improve focus. Clearly state the specific research objectives or hypotheses at the end of the introduction section.

4. Provide more detailed demographic information.

5. More information on how potential confounding variables  (e.g., physical activity, medication) were controlled would strengthen the study.

5. Table 1 is difficult to read.

6. Some figures could be simplified to enhance clarity. For instance, combining related data into fewer, more comprehensive visuals might make it easier to understand.

7. The chosen statistical methods align with the study's objectives. However, a justification for the sample size based on power calculations would strengthen the methodology.

7. Expand on the limitations and discuss how they could affect the findings.

8. Provide specific recommendations for future research, such as longitudinal studies or mechanistic investigations.

9. Highlight the practical implications of the study more explicitly in the Conclusion section.

Comments on the Quality of English Language

Minor editing of the English language is required.

Author Response

The authors very much appreciated the constructive comments on this manuscript by the reviewer. The comments have been very thorough and useful in improving the manuscript.

Comment 1: Highlight the implications or potential applications of the study’s findings in the abstract section.
Response 1: Corrected. We have made appropriate changes to the manuscript.

Comment 2: The review of literature could be expanded to include more diverse studies, particularly those exploring different dietary patterns and their specific impacts on the gut microbiome. Include a broader range of literature to provide a more comprehensive background.
Response 2: Corrected. We have added more information about the effects of different diets on microbiome composition.

Comment 3: The research objectives could be stated more clearly. While the introduction touches upon various aspects, a succinct statement of the study's aim would improve focus. Clearly state the specific research objectives or hypotheses at the end of the introduction section.
Response 3: Corrected. We have made appropriate changes to the manuscript.

Comment 4: Provide more detailed demographic information.
Response 4: Corrected. We have added additional information about the study participants.

Comment 5: More information on how potential confounding variables  (e.g., physical activity, medication) were controlled would strengthen the study. Table 1 is difficult to read.
Response 5: Corrected. We have added information on the physical activity level of the study participants. The most important medications that may have an effect on the composition of the gut microbiome are included in the exclusion criteria. Additional information was not collected. The Table 1 has been split into 2 and additional demographic information has been added.

Comment 6: Some figures could be simplified to enhance clarity. For instance, combining related data into fewer, more comprehensive visuals might make it easier to understand.
Response 6: We have considered your feedback and made the appropriate adjustments to the figures.

Comment 7: The chosen statistical methods align with the study's objectives. However, a justification for the sample size based on power calculations would strengthen the methodology.
Response 7: The objective of this study was not to examine the effects of drugs or dietary interventions. Additionally, the microbiome exhibits a high degree of variability, which makes it challenging to conduct power calculations. However, we will consider your valuable comments in future studies.

Comment 8: Expand on the limitations and discuss how they could affect the findings.
Response 8:  Corrected. We have added information about the limitations of this study to the discussion section.

Comment 9: Provide specific recommendations for future research, such as longitudinal studies or mechanistic investigations.
Response 9: Corrected. We have added possible suggestions for future research.

Comment 10: Highlight the practical implications of the study more explicitly in the Conclusion section.
Response 10: Corrected. We have made appropriate changes to the manuscript.

Reviewer 2 Report

Comments and Suggestions for Authors

The manuscript by Livantsova et al. provides valuable data on the effects of diet on gut microbiota in young obese adults.

The report deserves to be published, however, before this, the authors must improve its quality and address some concerns regarding bioinformatical data analysis.

1. Please, consider writing the terms "microbiota" and "microbiome" correctly. When you are showing the results of bacterial alpha and beta diversity assessment, as well as the bacterial differential abundance analysis, the term "microbiota" should be applied, as it represents bacterial communities. While the term "microbiome" represents bacterial communities and their "theatre of activity" (https://doi.org/10.1186/s40168-020-00875-0).

2. Figures. Please, avoid using prefixes, such as "f___", "g___", e.t.c. Their use is redundant, especially in the text of the manuscript. Also, please write taxa names in italics. For example, figure 3 contains names of the bacterial taxa not written in italics.

3.  L. 195. The authors state that they used the Knomics-Biota analytical system for the microbiota data analysis. However, the use of this system raises serious concerns.

3.1. First of all, ref. 24 is not the paper about Knomics-Biota software. Current ref. 24 is the paper about the feature-classifier plugin for QIIME2. The same mistake was made for ref. 25. Ref. 25 is the paper about the DADA2 denoising algorithm, which is also used for the creation of amplicon sequence variants data frames. Denoising is a completely different thing from quality-trimming and filtering (L. 196). Also, the authors already stated that they trimmed the data (L. 186). Was the data trimmed twice?

The authors have to revise the paper to provide correct references and a description of the methods used.

3.2. The authors do not state what method for the differential abundance analysis they used, which points to the fact that it was probably conducted with the Knomics-Biota software. It is important to note that the actual paper about Knomics-Biota (https://doi.org/10.1186/s13040-018-0187-3) does not contain any descriptions of the differential abundance analysis. I have checked the Knomics-Biota website, and this tool indeed provides some outcomes, which look like the results of the differential abundance analysis. However, the absence of a description of the methods implemented for the differential abundance analysis raises serious concerns regarding the robustness of the obtained results.

There is also no description (both in the presented study and the Knomic-Biota paper) of the methods implemented for the normalization of the microbiota data and/or the calculation of the relative abundances of identified taxa, as well as the methods for the type I error control.

Important is the fact that the paper about the Knomics-Biota was published in 2018, and this software is still under beta-testing after six years (website: Knomics-biota web-service is currently available under beta-testing conditions), which also raises serious questions about the validity of using this software for the data analysis, while there are a lot of other more reliable tools for the microbiota data analysis. There are some recent bioinformatical benchmarking studies about the differential abundance analysis methods, which show reliable software: https://doi.org/10.1038/s41467-022-28034-z, https://doi.org/10.1093/bib/bbac607, https://doi.org/10.1371/journal.pcbi.1010467.

I recommend the authors either provide sufficient information about the methods used for the differential abundance analysis, as well as the tests proving the robustness of the obtained results or conduct the analysis with reliable software.

4. L. 198-206. Alpha and beta diversity. The authors do not provide information about the rarefaction depth, although this is the most valuable information for the alpha and beta diversity analyses based on high-throughput 16S rRNA sequencing data. Also, the authors state that they used OTUs for the alpha and beta diversity analyses, which is not considered a reliable approach for the diversity analyses. Recent studies show that the use of amplicon sequence variants (ASVs) provides more robust results for diversity analyses in comparison to the OTU-based approach: https://doi.org/10.1371/journal.pone.0264443, https://doi.org/10.3390%2Fbioengineering9040146, https://doi.org/10.1186/s12864-020-07126-4.

Please, consider re-conducting microbiota data analysis using an ASVs-based approach.

Also, it is strange that the authors decided to use OTUs for the diversity analyses, while they used DADA2 for the prior analysis, which is also used for the generation of ASV data frames.

4.1. Also, the authors state that they used only the Shannon index in the M&M section, while they provide results of alpha diversity assessment with the Chao1 index (L. 295). The manuscript has to be carefully revised, so the M&Ms match the presented results. 

5. L. 337 and L. 348. The same goes for the Pearson correlation analysis. There is no mention of this statistical test in the M&M section. Also, the use of the Pearson test by itself raises questions regarding the expediency of its application in the search for the correlation between bacterial abundance and dietary macronutrient intake. There are various robust methods within the differential abundance approaches that allow for the inclusion of continuous data as the factors on par with nominal data (MaAsLin2, LinDA, etc.). Please, consider using other reliable approaches for the testing of the relationships between bacterial abundance and dietary macronutrient intake.

6. The authors state that they used high-throughput sequencing of one 16S rRNA region (V4) for the microbiota data analysis on the species level. V4 amplicons cannot be used for the analysis on the species level. Only full 16S rRNA amplicons can be used for the microbiota analysis on the species level. This has to be revised. The authors should not conduct analysis on the species level using only V4 16S rRNA data.

Conclusion: 

The manuscript provides valuable data for public health, and the study's topic with no doubts is significant. However, the data analysis report in its current form does not correspond to the current microbiota research criteria. I encourage the authors to conduct microbiota data analysis with reliable tools and provide robust results, which will improve the quality of the manuscript. 

Author Response

The authors very much appreciated the constructive comments on this manuscript by the reviewer. The comments have been very thorough and useful in improving the manuscript.

Comment 1: Please, consider writing the terms "microbiota" and "microbiome" correctly. When you are showing the results of bacterial alpha and beta diversity assessment, as well as the bacterial differential abundance analysis, the term "microbiota" should be applied, as it represents bacterial communities. While the term "microbiome" represents bacterial communities and their "theatre of activity" (https://doi.org/10.1186/s40168-020-00875-0).
Responce 1: Corrected. We have made appropriate changes to the manuscript.

Comment 2: Figures. Please, avoid using prefixes, such as "f___", "g___", e.t.c. Their use is redundant, especially in the text of the manuscript. Also, please write taxa names in italics. For example, figure 3 contains names of the bacterial taxa not written in italics.
Responce 2: Corrected. We have made appropriate changes to the manuscript.

Comment 3: L. 195. The authors state that they used the Knomics-Biota analytical system for the microbiota data analysis. However, the use of this system raises serious concerns.

3.1. First of all, ref. 24 is not the paper about Knomics-Biota software. Current ref. 24 is the paper about the feature-classifier plugin for QIIME2. The same mistake was made for ref. 25. Ref. 25 is the paper about the DADA2 denoising algorithm, which is also used for the creation of amplicon sequence variants data frames. Denoising is a completely different thing from quality-trimming and filtering (L. 196). Also, the authors already stated that they trimmed the data (L. 186). Was the data trimmed twice?

The authors have to revise the paper to provide correct references and a description of the methods used.

3.2. The authors do not state what method for the differential abundance analysis they used, which points to the fact that it was probably conducted with the Knomics-Biota software. It is important to note that the actual paper about Knomics-Biota (https://doi.org/10.1186/s13040-018-0187-3) does not contain any descriptions of the differential abundance analysis. I have checked the Knomics-Biota website, and this tool indeed provides some outcomes, which look like the results of the differential abundance analysis. However, the absence of a description of the methods implemented for the differential abundance analysis raises serious concerns regarding the robustness of the obtained results.

There is also no description (both in the presented study and the Knomic-Biota paper) of the methods implemented for the normalization of the microbiota data and/or the calculation of the relative abundances of identified taxa, as well as the methods for the type I error control.

Important is the fact that the paper about the Knomics-Biota was published in 2018, and this software is still under beta-testing after six years (website: Knomics-biota web-service is currently available under beta-testing conditions), which also raises serious questions about the validity of using this software for the data analysis, while there are a lot of other more reliable tools for the microbiota data analysis. There are some recent bioinformatical benchmarking studies about the differential abundance analysis methods, which show reliable software: https://doi.org/10.1038/s41467-022-28034-z, https://doi.org/10.1093/bib/bbac607, https://doi.org/10.1371/journal.pcbi.1010467.

I recommend the authors either provide sufficient information about the methods used for the differential abundance analysis, as well as the tests proving the robustness of the obtained results or conduct the analysis with reliable software.

Responce 3:  We have added a link to a publication describing the Knomics-Biota platform (https://doi.org/10.1186/s13040-018-0187-3). We also included a more detailed description of the microbiota analysis in the manuscript. The Knomics-Biota platform is an automated pipeline that includes standard analysis tools used to reconstruct microbiota composition. (https://doi.org/10.1016/j.csbj.2020.01.007, https://doi.org/10.3390/microorganisms907146, https://doi.org/10.1038/s41522-022-00342-8).

Comment 4:  L. 198-206. Alpha and beta diversity. The authors do not provide information about the rarefaction depth, although this is the most valuable information for the alpha and beta diversity analyses based on high-throughput 16S rRNA sequencing data. Also, the authors state that they used OTUs for the alpha and beta diversity analyses, which is not considered a reliable approach for the diversity analyses. Recent studies show that the use of amplicon sequence variants (ASVs) provides more robust results for diversity analyses in comparison to the OTU-based approach: https://doi.org/10.1371/journal.pone.0264443, https://doi.org/10.3390%2Fbioengineering9040146, https://doi.org/10.1186/s12864-020-07126-4.

Please, consider re-conducting microbiota data analysis using an ASVs-based approach.

Also, it is strange that the authors decided to use OTUs for the diversity analyses, while they used DADA2 for the prior analysis, which is also used for the generation of ASV data frames.

4.1. Also, the authors state that they used only the Shannon index in the M&M section, while they provide results of alpha diversity assessment with the Chao1 index (L. 295). The manuscript has to be carefully revised, so the M&Ms match the presented results. 

Responce 4:  Yes, you are right. We acknowledge that recent studies have advocated the use of ASVs rather than OTUs due to the potential biases introduced by the clustering methods used to delineate OTUs. However, the aim of our study was to investigate the main differences in the abundance of microorganisms of the gut microbiome in groups of individuals with disorders of carbohydrate and lipid metabolism, and the relationship with biochemical markers and nutrient intake. The use of OTUs, although a relatively outdated approach, is acceptable for this analysis (https://doi.org/10.1038/s41598-024-53837-z, https://doi.org/10.3390/nu14142993, https://doi.org/10.1155/2023/6868417). Unfortunately, we do not have the opportunity to analyze the microbiota data using an ASVs-based approach in a short period of time. However, we will consider your comments for future research.
4.1 Corrected. The mistake in the text of the manuscript has been corrected.

Comment 5: L. 337 and L. 348. The same goes for the Pearson correlation analysis. There is no mention of this statistical test in the M&M section. Also, the use of the Pearson test by itself raises questions regarding the expediency of its application in the search for the correlation between bacterial abundance and dietary macronutrient intake. There are various robust methods within the differential abundance approaches that allow for the inclusion of continuous data as the factors on par with nominal data (MaAsLin2, LinDA, etc.). Please, consider using other reliable approaches for the testing of the relationships between bacterial abundance and dietary macronutrient intake.
Responce 5:
Yes, you are right, using specialized statistical analysis techniques can have certain advantages. However, in this case, we were studying the relationship between nutrient intake and microbial abundance in a fairly homogeneous group of individuals. In general, the use of correlation analysis methods such as Pearson or Spearman is also acceptable and widely used in such studies (https://doi.org/10.1038/s41598-022-25041-4, https://doi.org/10.3389/fmicb.2021.712081,  https://doi.org/10.1007/s12072-021-10279-3) .

Comment 6: The authors state that they used high-throughput sequencing of one 16S rRNA region (V4) for the microbiota data analysis on the species level. V4 amplicons cannot be used for the analysis on the species level. Only full 16S rRNA amplicons can be used for the microbiota analysis on the species level. This has to be revised. The authors should not conduct analysis on the species level using only V4 16S rRNA data.
Responce 6: Yes, we agree, only full 16S rRNA amplicons can be used for the comprehensive microbiota analysis on the species level. However, our species-level microbiota analysis included only those species that were well differentiated by V4 amplicon. Other species are presented as unclassified. We have added relevant information to the study limitations section. (https://doi.org/10.1186/s40168-021-01168-w https://doi.org/10.3390/nu13103459, https://doi.org/10.3390/nu14091774, https://doi.org/10.1111/ijfs.15324).

Round 2

Reviewer 1 Report

Comments and Suggestions for Authors

Please mark the corrected sections in red in the revised manuscript.

Comments on the Quality of English Language

Moderate editing of English language required.

Author Response

Comment 1: Please mark the corrected sections in red in the revised manuscript.
Responce 1: Corrected. We have highlighted in color the major changes made to the manuscript.

During the re-checking of the results, we discovered some errors in the calculations and other inaccuracies and corrected them.  Corrections have been made to the results section (Figure 4) and discussion. Additionally, a calculation method has been added to the statistics section.

Reviewer 2 Report

Comments and Suggestions for Authors

The authors improved the manuscript, but still, they did not address all the comments properly.

Comment 1. (Responce 3)

1.1. Indeed, the authors expanded the description of the microbiota analysis in the manuscript, but still they did not address the issue regarding the differential abundance analysis. There is no description of the method used for the differential abundance analysis, but the authors report the results of this kind of analysis (e.g. Figure 4d). As the authors report log2fc, the method should be based on log2 transformation. Please, clarify what method was used for the differential abundance analysis. And also how the type I error was managed within the differential abundance analysis. This information is critical for the study's reproducibility. 

1.2. Please, write the specific QIIME2 version used for the data analysis (e.g. version 2023.5, 2023.7, etc..).

1.3. I cannot access the paper via the link provided by the authors (https://doi.org/10.3390/microorganisms907146). Indeed, Knomics-Biota software was used for the microbiota data analysis by some researchers. However, none of the papers provided by the authors contains a description of the differential abundance analysis methods.

Comment 2. (Responce 4)

Please, address the limitations of the OTU-based microbiota data analysis approach in the text of the manuscript, as you did with the V4 16S rRNA sequencing.

Comment 3.  (Responce 4.1)

The mistake in the text of the manuscript was not corrected. The M&M section still lacks mention of the Chao1 index.

Comment 4. (Responce 5)

Please, address the limitations of using Pearson correlation to test the relationship between bacterial abundance and dietary macronutrient intake, as you did with the V4 16S rRNA sequencing.

Comment 5. Please, provide references for the study limitations section. Provide the references on the studies, where the V4 16S rRNA region was used for the definitive species-level identification, in particular.

Comment 6. Data Availability Statement. Please, provide an accession number to the raw data uploaded to the public database (e.g. NCBI SRA). You can make the data publicly available after acceptance of the manuscript for publication.

Conclusion. 

Despite the improvements in the manuscript, the authors did not address important issues regarding microbiota data analysis. The manuscript needs to be majorly revised.

Author Response

The authors would like to thank you for your valuable comments. During the re-checking of the results, we discovered errors in the calculations and other inaccuracies and corrected them.

Comment 1.

1.1. Indeed, the authors expanded the description of the microbiota analysis in the manuscript, but still they did not address the issue regarding the differential abundance analysis. There is no description of the method used for the differential abundance analysis, but the authors report the results of this kind of analysis (e.g. Figure 4d). As the authors report log2fc, the method should be based on log2 transformation. Please, clarify what method was used for the differential abundance analysis. And also how the type I error was managed within the differential abundance analysis. This information is critical for the study's reproducibility. 

1.2. Please, write the specific QIIME2 version used for the data analysis (e.g. version 2023.5, 2023.7, etc..).

1.3. I cannot access the paper via the link provided by the authors (https://doi.org/10.3390/microorganisms907146). Indeed, Knomics-Biota software was used for the microbiota data analysis by some researchers. However, none of the papers provided by the authors contains a description of the differential abundance analysis methods. 

Response 1. 

  1. A number of errors were identified in the calculation of differences in the abundance of microorganisms between the studied groups. Corrections have been made to the results section (Figure 4) and discussion. Additionally, a calculation method has been added to the statistics section.
  2. We have updated the QIIME version.
  3. We apologize for the broken link (https://doi.org/10.3390/microorganisms9071461). 

Comment 2. (Responce 4)

Please, address the limitations of the OTU-based microbiota data analysis approach in the text of the manuscript, as you did with the V4 16S rRNA sequencing.

Response 2. 

Corrected. We have included relevant information in the study limitations section.

Comment 3.  

The mistake in the text of the manuscript was not corrected. The M&M section still lacks mention of the Chao1 index.

Response 3. 

Corrected. We have included both indices (Shannon and Chao1) in the M&M section.

Comment 4. 

Please, address the limitations of using Pearson correlation to test the relationship between bacterial abundance and dietary macronutrient intake, as you did with the V4 16S rRNA sequencing.

Response 4. 

Corrected. We have included relevant information in the study limitations section.

Comment 5. Please, provide references for the study limitations section. Provide the references on the studies, where the V4 16S rRNA region was used for the definitive species-level identification, in particular.

Responce 5. 

Corrected. We have included relevant information in the study limitations section.

Comment 6. Data Availability Statement. Please, provide an accession number to the raw data uploaded to the public database (e.g. NCBI SRA). You can make the data publicly available after acceptance of the manuscript for publication.

Response 6. 

Corrected. We have added the project to the NCBI SRA in the Data Availability Statement section (The data will be available after publication).

Round 3

Reviewer 1 Report

Comments and Suggestions for Authors

1. In the Introduction section, while the introduction discusses the general importance of the gut microbiome in obesity, it could benefit from a clearer identification of specific research gaps. Highlighting what previous studies have not addressed or where they have conflicting results would strengthen the rationale for this study. The objectives are mentioned but could be more explicitly outlined in a separate section or paragraph to enhance clarity. Explicitly stating the hypotheses could also be beneficial.
2. Maintain consistency in terminology and style throughout the sections to provide a seamless reading experience.

Comments on the Quality of English Language

Ensure that the language used is clear and concise throughout the paper. Avoid overly complex sentences and jargon that could hinder comprehension.

Author Response

Comment 1:  In the Introduction section, while the introduction discusses the general importance of the gut microbiome in obesity, it could benefit from a clearer identification of specific research gaps. Highlighting what previous studies have not addressed or where they have conflicting results would strengthen the rationale for this study. The objectives are mentioned but could be more explicitly outlined in a separate section or paragraph to enhance clarity. Explicitly stating the hypotheses could also be beneficial.

Responce 1: Corrected. We have added information about modern areas of research as well as some gaps remaining. We rewrite the aims to clarify it.

Comment 2: Maintain consistency in terminology and style throughout the sections to provide a seamless reading experience.

Response 2: We have made a number of changes and fixed some questionable wording for more academic text style.

Reviewer 2 Report

Comments and Suggestions for Authors

The authors addressed all the issues.

Author Response

No comments required.